# Elemental Screening and Nutritional Strategies of Gypsophile Flora in Sicily

**DOI:** 10.3390/plants14050804

**Published:** 2025-03-05

**Authors:** Antonio J. Mendoza-Fernández, Encarna Merlo, Carmelo M. Musarella, Esteban Salmerón-Sánchez, Fabián Martínez-Hernández, Francisco J. Pérez-García, Giovanni Spampinato, Juan Mota

**Affiliations:** 1Department of Botany, University of Granada, 18071 Granada, Spain; 2Department of Biology and Geology, CEI·MAR, CECOUAL, ENGLOBA, University of Almeria, 04120 Almeria, Spain; emerlo@ual.es (E.M.); esanchez@ual.es (E.S.-S.); fmh177@ual.es (F.M.-H.); fpgarcia@ual.es (F.J.P.-G.); jmota@ual.es (J.M.); 3Department of AGRARIA, Mediterranean University of Reggio Calabria, 89122 Reggio Calabria, Italy; carmelo.musarella@unirc.it (C.M.M.); gspampinato@unirc.it (G.S.)

**Keywords:** BCF (bioconcentration factor), gypsophyte, gypsophily, gypsophilous flora, ionome, chemical composition, Mediterranean gypsum outcrops, nutrients

## Abstract

Sicily is a Mediterranean island with an exceptional natural heritage, where gypsum outcrops are widespread and associated with an endemic flora. These ecosystems are prioritized by the European Habitats Directive (Mediterranean gypsum steppes, 1520*) in the Mediterranean Basin. Some studies have revealed the physiological mechanisms in gypsophile plants, which are important adaptative characteristics of plants that live on gypsum. To identify stress-tolerant strategies, we studied the leaf chemical composition of 14 plant species (gypsum endemics, Mediterranean gypsophiles and widely distributed) from Sicily. The ability to accumulate mineral elements in leaves, especially sulfur (S), calcium (Ca) and magnesium (Mg), is a widespread strategy for gypsophile plants. Bioconcentration factor (BCF) calculations also indicate bioaccumulation of carbon (C), nitrogen (N), and potassium (K) in species with a certain degree of foliar succulence, such as *Gypsophila arrostii* Guss. subsp. *arrostii* or *Diplotaxis harra* (Forssk.) Boiss. subsp. *crassifolia* (Raf.) Maire, which also accumulates Mg and Sodium (Na). The narrow gypsophile *Erysimum metlesicsii* Polatschek exhibited the highest BCF value for strontium (Sr). The study of the gypsophile *G. arrostii* subsp. *arrostii* growing on limestone substrates indicates that this plant tends to hyperaccumulate nutrients, such as S, that are normally available in gypsum substrates. The remarkable ability of these plants to absorb elements such as sulfur and strontium is important to explain their ecological adaptations but also indicates their potential usefulness in environmental phytoremediation processes. The study of plant communities and flora of gypsum substrates is essential to understand the nutritional adaptations that allow flora to survive in gypsum environments and to support the better preservation of these interesting natural areas in Sicily.

## 1. Introduction

The term edaphism refers to ‘geobotanical phenomenon giving rise to particular floras on certain substrates’ [1] or ‘those physical and chemical effects induced on living beings by the soil’ [2] and has been used extensively in Europe since the 19th century [3,4,5,6], as well as in the increasing research in this topic [7]. Andrea Cesalpino in *De plantis libri XVI* [8], which is considered the first textbook of botany, documented the existence of endemic plant species on the Italian serpentines. In addition, gypsum substrates are a clear example of the link established between the flora and substrate. Specifically, Parsons [9] defined the concept of plant gypsophily as plants’ exclusive (or almost) preference to live on gypsum substrates. Thus, plant species that grow exclusively in gypsum outcrops are called gypsophytes [10]. In a certain way, these substrates could act as very selective habitats for flora due to their harsh conditions, which are, in addition, determined and modeled by the region in which each outcrop occurs. In this regard, the investigations conducted by Pérez-García et al. [11,12] confirm the widespread dissemination of this phenomenon worldwide.

In Italy, the existence of a characteristic flora living on gypsum substrates was already evidenced by the botanist L. Macchiati at the end of the 19th century [13,14,15]. Further studies to characterize the vegetation in Italian gypsum outcrops expanded the knowledge about this flora in mainland Italy [16,17,18,19,20,21,22,23,24], and more deeply in some regions located in Sicily [25,26,27,28,29,30,31,32]. These plant communities have been included on the European Red List of Habitats [33] and are considered priority habitats (Mediterranean gypsum steppes *1520) for conservation according to 92/43 EU Directive [34]. The aforementioned idea that only plant species showing a preference—or even a complete preference—for gypsum substrates would be classified as gypsophytes is a very complex topic still under study [10,12,24,35,36,37,38,39,40]. In their review, Mota et al. [41] recognized alternative approaches and discussed their use to elucidate which species could be considered gypsophytes. One of these approaches would be to establish the existence of several adaptive strategies that can be recognized in the gypsophilous flora [42], which could be the most parsimonious explanation. Among these strategies, the most frequently mentioned ones relate to the nutritional restrictions of gypsum soils for plant growth [43,44,45] or the accumulation of certain elements in their tissues, mainly foliar [38,46,47,48,49]. Gypsum soils contain high concentrations of calcium (Ca), magnesium (Mg), and sulfur (S), which can also be accumulated in the tissues of the plants inhabiting there; therefore, they are considered key elements in the interpretation of gypsophily [38]. Nevertheless, these data were unavailable for most gypsophilous species [37,42], and even less in territories where little attention has been paid to them from a functional point of view. It is remarkable that not all plant species living on gypsum adopt a single strategy [37,50]. Some studies have revealed certain mechanisms in gypsophile plants, which are an important part of plant adaptation to live on gypsum. Studies have also revealed that soil elemental composition (e.g., high Ca, Mg and S concentrations, low cation exchange capacity) may control community structure on gypsum substrates [38], so foliar analyses provide quite an accurate indication of the absorption of soil elements by plants [51]. Henceforth, research works strongly supported by detailed empirical data could help to clarify the role played by the gypsum soil on the nutritional characteristics of gypsicolous flora, as has been performed in other parts of the world with other gypsophile species [52,53,54]. This type of approach has also achieved interesting results in other environments, such as dolomitic soils, where imbalances between the proportions of elements such as Ca and Mg seem to be determinants of plant community development [55].

The main objective of this research was to identify nutritional strategies for the gypsophile flora among the species present in the gypsum outcrops of Sicily and to determine any ecological strategies that are stress-tolerant for these plants. Second, we compared the nutritional strategies to previous classifications of plant functional types [24,38,40,50]. Finally, we investigate the peculiarities of the Sicilian gypsum outcrop flora to support their relevance as European Priority Habitats for conservation.

## 2. Results and Discussion

### 2.1. Soil Analyses

Gypsum contents in the sampled soils varied considerably from percentages above 95% in the Rocca di Entella outcrop to the lowest value registered in the collected soil from Castelmola locality, 2.96% (Table 1), which could not be considered as a gypsiferous soil itself. Except for this last result, the sampled soils showed gypsum concentrations higher than 50% on average. Regarding the gypsum concentration, van Alphen and de los Rios Moreno [56] used the term ‘gypsiferous soils’ to refer to those soils containing more than 2% gypsum. Following this threshold, even the soil sample from Castelmola with 2.96% gypsum content could be considered a slightly gypsicolous substrate. However, other authors [57,58,59] postulated a concentration of 25% in gypsum as a reference value to recognize soil limitations related to water-holding capacity, ion imbalance, or physical surface crust presence. Mardoud [60] concluded that it is very difficult or impossible for the roots of forest species to penetrate layers with more than 60% gypsum concentrations. The maximum gypsum contents in the soil samples (>70%) can be related to the highest S concentrations (3.92 g 100 g^−1^ in Monte Gibliscemi, or 3.22 g 100 g^−1^ in Rocca di Entella gypsum outcrops) (Table 1 and Table 2). Moreover, these high contents produce remarkably unbalanced soils, which may cause effects on the biogeochemical composition or imbalances of some relevant element ratios in plant tissues [36]. The electrical conductivity analysis results indicated that all samples showed low electrical conductivity, as previously indicated for gypsum soils [61,62,63]. Thus, they can be classified as non-saline (<2 dS m^−1^) to slightly saline soils (2 to <4 dS m^−1^) [64]. In general, they were slightly alkaline, with an average pH of 7.8. Gypsisols in arid zones have electrical conductivity values that usually vary between 2 and 8 dS m^−1^, and pH values between 7.4 and 8.4 [65]; thus, the results can be considered within the usual ranges in both cases. As demonstrated by several studies [51,66,67,68,69], electrical conductivity and pH are strongly correlated with Mg and Na content in soils, which can affect the availability of nutrients to plants. In the sampled soils, the Mg content did not exceed 0.52 g 100 g^−1^ and Na content 0.04 g 100 g^−1^, showing an inverse relationship between the percent gypsum and Na concentration.

The N and P contents in the soils were low. As pointed out by Vitousek et al. [70], these elements are the most common limiting nutrients in a wide variety of terrestrial ecosystems. Conversely, the Sr concentrations were high, especially at the Rocca di Entella locality, where the highest percentages of gypsum content were also found (Table 2). These results are consistent with the data published by Merlo et al. [38], who reported the common occurrence of high Sr concentrations in gypsum soils.

### 2.2. Foliar Analyses

Table 3 presents the average values obtained in the leaf analyses for the macronutrients in % (g 100 g^−1^) and for the ratio N:P, considered to be of interest.

#### 2.2.1. C Contents

The average C concentration of the studied species was around 40 g 100 g^−1^, which could be considered the standard level [38]. The lowest values of the group evaluated were those of *D. harra* subsp. *crassifolia*, coinciding with previously published data by Merlo et al. [38]. Low C content is associated with low structural complexity of leaves or less succulency [71]. Conversely, elevated C values exceeding 55 g 100 g^−1^ were found in the shrub species *Erica multiflora* subsp. *multiflora* (Table 3). Furthermore, *Brassica villosa* subsp. *tineoi* and *S. gypsicola* subsp. *trinacriae*, representing narrow and wide gypsophile species, respectively, had slightly higher C content leaves. The large amount of specific secondary compounds in Brassicaceae species or CAM metabolism in Crassulaceae plant species could probably explain their C content [72].

#### 2.2.2. N Contents

The group of sampled plants seemed to obtain N efficiently, except in the case of species belonging to the Crassulaceae family, which presented minimum values (Table 3). In general, according to Merlo et al. [38], N contents can be considered moderate or low in gypsophytes. However, N content has been used as a criterion to distinguish ‘wide gypsophile’ species. Palacio et al. [50] suggested that the concentration of N in leaf samples could be understood as a clear characteristic to define a gypsophilous plant (in Spain, it has been used to catalog the plant species *Lepidium subulatum* L. as a ’wide gypsophile’). On the other hand, *G. arrostii* (Cariophyllaceae) presented high N concentrations on average. Some of the most common plants on gypsum around the world belong to *the Gypsophila* genus [11,12], and their success could be due to the efficiency of obtaining nitrogen in oligotrophic soils [45]. The ability to assimilate this nutrient confers an adaptive advantage that can be found on a global scale and in other environments, as in the case of dolomitic soils [36,55,73]. Furthermore, the Asteraceae specimen (*A. arborescens*) was adjusted to this pattern.

#### 2.2.3. N:P Ratio

The N:P ratio has been demonstrated to be useful for analyzing the effects of these nutrient deficiencies in plants [74,75]. Most values of the studied plants from gypsum soils in Sicily were between 10 and 25, which is assumed to be regular levels according to the literature [38,76,77,78,79]. Although almost none of the studied plants had a N deficit, Table 4 shows that the N:P ratio (when values > 20) indicates a probable deficiency in P concentrations [80]. Merlo et al. [37] and Drohan and Merkler [81] reported that P is the scarcest nutrient for plant species living on gypsum soils. This conclusion matched the average foliar P content, which was lower than 0.19 g 100 g^−1^ in the sampled gypsophytes. P could be the most restrictive macronutrient for plants growing on gypsum substrates, and this fact coincides with the insufficient availability of this element in these soils [82]. Moreover, P is mainly present as HPO_4_ ions at pH > 7.0, which are difficult for plant roots to absorb [56]. This finding is particularly interesting because the deficit of essential nutrients, such as P and N, has been used as an important explanatory factor for gypsophily in plants [83,84,85], and despite this fact, some plants are able to grow on gypsum soils.

#### 2.2.4. Ca Contents

Gypsophilous soils are rich in Ca and relatively poor in Mg [36]. Among plants growing on gypsum, there are some well-known Ca accumulator plants [37]. Certainly, very high Ca contents (>3.5 g 100 g^−1^) have been found in narrow gypsophytes such as *G. arrostii*, the Cariophyllaceae family. However, Crassulaceae species (Figure 1) also showed high Ca concentrations in leaves, and the gypsovag *P. sediforme* reached the maximum value for Ca in this study (Table 4). The characteristic succulence of the leaves of *Sedum* and *Petrosedum* genera could be used to determine the accumulation of this element in foliar tissues [41]. This could be related to the water economy of plants through the maintenance of the osmotic balance [38]. Ca concentrations could also be understood as anti-herbivore protection [36,86,87]. In fact, the existence of Ca crystals has been established in the tissues of plants living on gypsum [62,88]. On the other hand, according to the threshold values proposed by Merlo et al. [38], *E. multiflora* subsp. *multiflora*, *A. arborescens*, *B. villosa* subsp. *tineoi*, and *S. fruticosa* could be considered plants deficient in Ca. This is due to the fact that they maintain low contents of this nutrient despite it being available in the environment, which would indicate a possible exclusion strategy, while *G. arrostii* and *P. sediforme* showed the highest values. This is interesting because it indicates their possible strategy as accumulator plants. Finally, the remaining species had regular values.

#### 2.2.5. S Contents

Some studied gypsophytes had S contents clearly above the average. Despite the fact that *E. multiflora* subsp. *multiflora*, *S. caeruleum*, *S. gypsicola* subsp. *trinacriae*, and *S. fruticosa* could be considered S-deficient species (non-accumulator species), *P. ochroleucum* subsp. *mediterraneum*, *P. sediforme*, *A. arborescens*, *E. metlesicsii*, *P. illyrica* subsp. *angustifolia*, *B. villosa* subsp. *tineoi* presented values estimated as typical, and *G. arrostii* and *D. harra* subsp. *crassifolia* were species that showed values of S above the “very high” threshold proposed by Merlo et al. [38]. The Cariophyllaceae family (Figure 1) presents high S concentrations in general (accumulator species). In addition, this ability is very patent in some species of the Brassicaceae family: *D. harra* subsp. *crassifolia*, *E. metlesicsii*, and *M. fruticulosa* subsp. *fruticulosa* (Figure 1). These significant concentrations of S may be related to their capacity to assimilate and metabolize excess S through the synthesis of secondary S compounds [89]. However, the results show that not all studied plants can accumulate S despite living on gypsum substrates, which are very rich in this element [90]. In the case of the Crassulaceae family species, the S content did not exceed 0.27 g 100 g^−1^ in any of the species, and the minimum values for the S concentrations were found in this gypsophyte group. As Merlo et al. [38] indicated, Ca accumulates more frequently than S among plants living on gypsum; this is especially evident in the Brassicaceae family (Table 4). No very high concentrations of S were found in the edaphic samples, probably because it is a relatively rainy area within the context of the Mediterranean Basin.

#### 2.2.6. Mg Contents

As Merlo et al. [38] indicated, Mg is a key element for understanding the gypsophily phenomenon. Some succulent-leaved gypsophytes have high levels of this element [62,91]. In the gypsum soils of Sicily, the highest Mg contents were found in species with a certain degree of succulence (*G. arrostii* or *D. harra* subsp. *crassifolia*). These results are consistent with the ability of certain species of the Caryophyllaceae family to accumulate Mg in the cell vacuole [92]. Nevertheless, in other succulent-leaved plants (*Sedum* and *Petrosedum* genera), Mg contents were lower. In this case, Crassulaceae family species presented higher Ca concentrations, close to the maximum values; conversely, Mg levels did not show the same results (Figure 1). Not surprisingly, the highest magnesium levels were recorded in the leaves of *P. illyrica* subsp. *angustifolia* (Cariophyllaceae). In addition, high Ca contents were found in this species. The reason could be that this plant was collected in Castelmola on a limestone substrate where the lowest gypsum content among all the studied soils was observed (2.96%). Following the threshold classification proposed by Merlo et al. [38], almost all the studied species revealed a Mg deficit, except *for G. arrostii*, *M. fruticulosa* subsp. *fruticulosa*, *D. harra* subsp. *crassifolia*, and *S. fruticosa* with normal values, and *P. illyrica* subsp. *angustifolia* with a high value, as already discussed above.

#### 2.2.7. Other Elements: Al, Fe and Sr

There are other interesting results related to the three elements. Species belonging to the Brassicaceae and Crassulaceae families showed some maximum Al and Fe concentrations. These results, well above the average values of 80 g 100 g^−1^ indicated by Merlo et al. [38], reached values above 700 g 100 g^−1^ in some cases of the Crassulaceae family. In addition, the correlation between high Fe and Al contents could be related either to P deficiencies or the formation of P complexes [38,93,94]. Gil and Ramos-Miras in Mota et al. [10] pointed to the low assimilable P content found in gypsum soils from Spain; thus, data from Sicilian soil analyses would support the first of the aforementioned hypotheses. A similar trend was observed for the Sr data. The endemic plant species *Erysimum metlesicsii* yielded values higher than 1400 mg Kg^−1^. In addition, other species belonging to the Brassicaceae family, such as *D. harra* subsp. *crassifolia* and *M. fruticulosa* subsp. *fruticulosa*, showed values above 1000 mg Kg^−1^ (*P. sediforme* also surpassed this value). These results indicate that these plants could be classified as Sr hyperaccumulators [38]. Therefore, the last two species mentioned could be useful in phytoremediation [36,38], since they are widely distributed and not threatened [24,95], and *E. metlesicsii*, with a restricted distribution, could act, at least, as a sentinel (or indicator) species.

### 2.3. Bioconcentration Factors (BCFs)

The calculated BCFs (Table 4) for the group of macroelements C, N, P, and K were noteworthy because the values were higher than one in all the cases studied. It was observed that plants tended to accumulate these elements in large quantities, reaching the highest average value of all elements (15.70 in the case of N). The gypsophilous plants from the Roca di Entella outcrop and the Castelmola locality presented the highest values in terms of the accumulation of this element. This result may be related to the fact that the soils of these two localities had the lowest N content. As already indicated, the species of the Crassulaceae family presented the highest values of Ca in leaves; however, with respect to the concentrations of this element in the soil, the only species that seemed to act as a Ca accumulator was *P. sediforme*. On the other hand, it was the species with the largest size of the Crassulaceae family taxa studied. Regarding Mg accumulation, the plants with BFC values > 1 were *G. arrostii*, *M. fruticulosa* subsp. *fruticulosa*, *D. harra* subsp. *crassifolia*, and the gypsovag *P. illyrica* subsp. *angustifolia*. B was a micronutrient that showed bioaccumulation in practically all the plants sampled.

Regarding this element, *Gypsophila sphaerocephala* Fenzl ex Tchihat. has been documented as a plant that can tolerate high levels of B in soil growing on gypsum substrates [96]. With respect to the BCFs obtained for S, while in the samples taken from the gypsum outcrops, the average BCF for S was 0.32, it was 11.55 in the samples from Castelmola. In other words, the BCFs for S of all species from the Castelmola locality were higher by as much as two orders of magnitude than those of the other species studied. Therefore, S does not appear to be an element that gypsophile species can bioaccumulate when it is sufficiently available in the soil, not even narrow gypsophiles such as *G. arrostii*. However, even when S content is low and the percentage of gypsum in the soil is not sufficient to be considered a gypsicolous soil (e.g., Castelmola: S: 0.13 g 100 g^−1^; gypsum%: 2.96%), the BCFs indicate that plants tend to accumulate this element, reaching analogous concentrations in leaves to those of plants growing on gypsum substrates. In Castelmola, the narrow gypsophile *G. arrostii* behaves as a S accumulator and also accumulates N, showing the highest value obtained for all plants (36.79). These findings may be associated with the primocolonizing character of the genus *Gypsophila* L. species, as demonstrated in studies on plant succession in gypsum quarries [97,98].

As Merlo et al. [38] highlighted, gypsophytes have very low Na contents. However, according to the BCF results, the studied taxa of the Brassicaceae family act as Na accumulators, although all the samples were obtained from soil with low electrical conductivity and thus low NaCl contents. *D. harra* subsp. *crassifolia* (average foliar Na contents 0.59 g 100 g^−1^), a species that also lives in subsaline environments, presented a similar behavior to *Diplotaxis harra* subsp. *lagascana* (DC.) O. Bolòs & Vigo (Na contents 0.69 g 100 g^−1^) from Spanish gypsum outcrops. The indices obtained for Sr were also noteworthy. The endemic *E. metlesicsii* can be considered a Sr accumulator, with a leaf content of 1416.57 mg kg^−1^ and a BCF of 3.77. In addition, *D. harra* subsp. *crassifolia* and *M. fruticulosa* subsp. *fruticulosa* had BCF values > 3.00. *P. sediforme* was the only species that reached this BCF score, although other species of the Crassulaceae family, such as *S. gypsicola* and *P. ochroleucum*, showed values above 1.00.

### 2.4. Statistical Tests

#### 2.4.1. ANOVA

ANOVA revealed significant differences between the nutritional element contents in leaf samples. In the case of taxonomical families, groups of 12 nutrients showed significant differences between groups, depending on the type of post hoc contrasts performed. Scheffe’s test revealed that the Cariophyllaceae family had significantly higher leaf content of C, N, Ca, S, B, Cu, Ni, and Sr than the other groups. Cariophyllaceae species are distinguished from Crassulaceae species in terms of concentrations of nutrients such as S, N, Mn, and B; from the Asteraceae family in terms of C, Ca, and Ni contents; and from Brassicaceae species in terms of Sr values. According to functional type, eight variables differed significantly between groups. Wide gypsophiles had high concentrations of leaf S and Ca, as well as other inorganic elements such as Mg, P (relatively low contents), and Na, compared to narrow gypsophiles and gypsovags. These findings are consistent with those of previous studies [38,88,99]. However, none of the multiple comparisons (post hoc Sheffe’s tests) for leaf chemical values between the wide and narrow gypsophile groups were significantly different, except in the case of Li content. The narrow gypsophile species showed more dissimilar leaf compositions to the gypsovag group; there were significant differences in five foliar mineral nutrients, including C, S, Al, Fe, and Li. Mean leaf S content of narrow gypsophiles exhibited the most marked differences among the aforementioned nutrients. Grouping by succulence character revealed several significant differences between leaf C, Ca, and S content. The succulent species had average leaf carbon contents lower than 40 g 100 g^−1^, which in addition coincided with those taxa that accumulated Ca. *G. arrostii*, *D. harra* subsp. *crassifolia*, and *P. sediforme* were species that strongly matched this trend. The succulence could be related to the water economy of plants through the maintenance of the osmotic balance [38]. High Ca concentrations can also be stored as a component of cell structures [100] and used as an anti-herbivore defense [36,62,86,87,88] (for further information about the statistical analyses performed, see Appendix A).

The study of the narrow gypsophile *G. arrostii* growing on limestone substrates indicates that this plant tends to hyperaccumulate nutrients that are normally available in gypsum substrates; in fact, ANOVA statistical analyses showed significant differences in 10 of the BCF results for the elements compared (N, Al, B, Cu, Fe, K, Li, Na, Rb, S). The most striking differences in the bioaccumulation of elements were found in the N and S contents, where the values obtained in plants growing on the limestone substrate were much higher (Table 5). In contrast, BCFs for Na and K showed indications of bioaccumulation, mainly in plants growing on gypsum.

#### 2.4.2. PCA and Terniary Plots

Figure 2 presents an ordering of the plant species by means of principal component analysis (Pearson coefficient). S, Ca, and Mg contents drove the separation of leaf chemical signatures among three clearly differentiated groups. As pointed out by Palacio et al. [40], elements like S, Mg and Ca show a significant increase in the variation explained by the phylogeny of gypsophile plants, as the three elements critical for plant adaptation to gypsum soils.

It can be observed how the species of the *Gypsophila* genus (Cariophyllaceae) acted as good accumulators of Ca and S, as indicated by individual results. Species of the Crassulaceae family (i.e., *S. gypsicola* subsp. *trinacriae*, *P. sediforme*, *P. ochroleucum* subsp. *mediterraneum*) had the highest Ca contents. However, the species of the Brassicacae family showed an opposing trend and had higher S concentrations than the other studied species, highlighting *D. harra* subsp. *crassifolia* with the highest S contents. In general, leaf concentrations of S and Ca were higher in wide gypsophiles than in narrowly distributed gypsophiles and gypsovags.

The ternary graphs presented in Figure 3 show, on the one hand, the generalized P limitation of plants growing on gypsum (Figure 3a) and, on the other hand, that the high proportion of S (Figure 3b) can hinder the absorption of the mentioned nutrient (P) [101]. In the plants studied, Mg values ranged from 0.06 to 0.93 g 100 g^−1^ for all species, and on average, they were five and ten times lower than the Ca and S contents, respectively. These results reinforce the physiological and molecular evidence of Mg homeostasis in plant cells [102]. They also suggest adaptations such as those proposed by Tyndall and Hull [103], such as a higher tolerance to low and high concentrations of Mg, a higher Mg requirement for maximum growth, and mechanisms to increase Mg absorption. The compartmentalization of ions, including Mg, in cellular organelles and in the apoplast of the cells of the tissues of the roots and leaves, mainly in the vacuoles, should be added to these mechanisms [55]. Furthermore, Mg increases the transport of sugars from the leaves to the roots, promoting the growth of the latter, modifying the root/stem ratio and increasing the water absorption surface [104,105,106].

On the other hand, *G. arrostii* appears to be the plant species that most efficiently manages N and Ca imbalances. As shown in the ternary plots (Figure 3), Mg, and to a lesser extent P, could be the most limiting nutritional elements for the studied species on gypsum, as already pointed out by Merlo et al. [38] in the case of the gypsophile flora growing in Spain. In addition, Muller et al. [107] demonstrated that widely distributed gypsum endemics from the Chihuahuan Desert (North America) displayed higher foliar S and higher whole-plant Mg than their non-endemic relatives, as occurred in the Iberian Peninsula. Plant species in the Crassulaceae family had the highest Ca contents in leaves; however, the lowest proportions of S content were also recorded in this group of species. Ca excess may be stored in crystalline phases such as calcium oxalate, a kind of crystal commonly found in gypsophile plants [88,91]. Although its role remains controversial, they may be related to Ca excess in soils [100].

## 3. Materials and Methods

### 3.1. Study Area

Sicily, located in the Mediterranean Basin, is a continental island considered an essential portion of the Mediterranean Hotspot [108,109] due to its endemic plants and exceptional biodiversity. While gypsum outcrops in Italy occur underground or scattered throughout the mainland along the whole Apennine-Maghrebide Mountain range [110] (Emilia Romagna is the widest gypsum outcrop, with an extent greater than 100 km^2^ [111]), in Sicily, these substrates cover an area greater than 1000 km^2^, which is the largest extent in Italy [112,113,114]. In terms of geology, the age of these outcrops ranged from the Paleozoic to the Tertiary period [110]. Although it does not always refer to pure gypsum outcrops, the geological group called ‘Serie Gessoso-Solfifera’ [115] includes different evaporitic facies that determine a great lithological heterogeneity, depending on the age of its genesis (Miocene period; Messinian, 7.2–5.3 Ma) [112,116]. They are distributed mainly in the Mediterranean bioclimate, with a lower semiarid or semiarid ombrotype [117].

A large part of the gypsum outcrops in Sicily are excellently studied and well documented. Some of these have been considered Sites of Community Importance, e.g., Complessi Gessosi Monte Conca (SCI ITA050006), where site-specific territory management plans have been tailored. However, in contrast to the conservation interest of the biodiversity heritage of these gypsum outcrops, it should be noted that this mineral constitutes a mining resource [118]. In Italy, which is the second largest gypsum producer in Europe (4100 kt year^−1^ [119]), the main gypsum outcrops were documented in detail in the middle of the 20th century to incorporate the use of this raw material in building processes, as happened in other European countries [110].

### 3.2. Fieldwork

Fieldwork conducted in Sicily occurred in spring (April to early May of the years 2009, 2013 and 2015) to obtain plant samples during the most favorable phenological period. Two collections of plant vouchers were deposited in the official herbaria of the Universita degli Studi ‘Mediterranea’ of Reggio Calabria (REGGIO) and in the University of Almeria one (HUAL).

The fieldwork consisted of sampling a representative group of flora species growing on gypsum outcrops in Sicily to collect leaf and soil samples. This process did not damage any flora specimens because the amount of tissue needed for the analyses was less than 10 g. Five replicates were performed per plant. Furthermore, soil samples were collected from the plants. The field data were georeferenced by GPS (GPSMAP 60CX equip, 2 m error, Garmin Ltd., Spain) and uploaded into a Geographic Information System [120].

### 3.3. Selected Plants

Leaf samples were collected from 14 plant species belonging to five taxonomical families (Table 6) from four different localities in Sicily. Due to the high β-diversity found in gypsum outcrops, not all taxa are present in every outcrop [118]. Consequently, flora species and localities were selected by considering both the distance between different populations and their presence/absence and abundance in each sampling location. Seven gypsophytes were included to cover the widest possible spectrum in terms of functional typology. Three of these were Italian gypsum endemics (narrow gypsophiles) abundant in the studied territory: *Brassica villosa* subsp. *tineoi* (Lojac.) Raimondo and Mazzola; *Erysimum metlesicsii* Polatschek; *Gypsophila arrostii* Guss. subsp. *arrostii* (*G. arrosti*). Four were Mediterranean distributed gypsophiles: (*Diplotaxis harra* (Forssk.) Boiss. subsp. *crassifolia* (Raf.) Maire; *Sedum gypsicola* Boiss. and Reut. subsp. *trinacriae* Afferni; *Petrosedum ochroleucum* (Chaix) Niederle subsp. *mediterraneum* (L. Gallo) Niederle; *Matthiola fruticulosa* (L.) Maire subsp. *fruticulosa*, which is widely distributed throughout Italy (wide gypsophiles). The remaining seven species were considered as gypsovags [24].

Only healthy green leaves were prepared to ensure that all samples had homogeneous phenological status. The botanical nomenclature was based on that used by Bartolucci et al. [95]. The collected foliar samples were transported to the laboratory, where they were washed with deionized water, dried at 70 °C until their weight stabilized, and finely grounded in a Restch Mortar Grinder, model MM 200 (Retsch GmbH, Haan, Germany).

### 3.4. Soil Samples

The edaphic test involved sampling representative gypseous places (the main environmental features of the sampled sites are listed in Table 2). For each edaphic sample, three subsampling pits were excavated in a square of 100 m^2^ (bushes and bedrock were avoided). The samples were collected from superficial soils (15–25 cm deep, where conditions allowed). For the subsamples, a minimum of 1 kg of material was taken and mixed and kept in labeled plastic bags. In the laboratory, the soil was spread out for drying and sieved to 2 mm for subsequent analysis [63]. Gypsum soil samples were paired with foliar samples of the selected plants to determine the elemental concentrations, average gypsum contents (% gypsum), electrical conductivity data (EC dS m^−1^) and pH in soils. The gypsum concentration in soils was determined using the method of Artieda et al. [121]. Electrical conductivity (EC) and soil pH were determined in saturated soil paste extract, using the corresponding glass electrode (Crison conductivimeter and Crison micropH 2001) [122].

### 3.5. Elemental Analysis

Both foliar and soil samples were analyzed in specialized external laboratories. Measurements were performed according to the CEBAS Ionomics Service methodology (https://www.cebas.csic.es). Total C and N analysis was performed using an Elemental Analyzer. In addition, 31 further elements were determined (Al, As, Be, Bi, B, Ca, Cd, Co, Cr, Cu, Fe, K, La, Li, Mg, Mn, Mo, Na, Ni, Pb, P, Rb, Sb, Se, Si, S, Sr, Ti, Tl, V, Zn) by inductively coupled plasma-optical emission spectrometry (ICP-OES). The foliar and soil contents of the macroelements are expressed in percentages (g 100 g^−1^), and the microelements are expressed in mg kg^−1^ of dry weight (DW) in both cases.

### 3.6. Bioconcentration Factor

The index quantifies the capability of a plant species to accumulate certain elements from soil at the tissue level [123]. Brooks [124] described this factor as the ratio of the concentrations of the element in question in plant material and soil, calculated using the following equation [125].(1)$$BCF=\frac{Concentration\;of\;elements\;in\;sprout\;tissue\;(\frac{mg}{kg})}{Soil\;element\;concentration\;(\frac{mg}{kg})}$$

Equation (1) is used for calculating the bioconcentration factor (BCF).

According to Buscaroli [126], plant species with BCF values > 1 have high accumulation potential, and they can even be used in phytoremediation techniques. The greatest accumulation of elements usually occurs in the aerial organs of plants [127]. Elevated element contents in soil but BCF < 1 could indicate the existence of other adaptative physiological processes, such as exclusion mechanisms.

Data from the localities of Monte Gibliscemi and Rocca di Entella were used to calculate the Bioconcentration Factor (BCF), as these outcrops were the ones with the highest percentage of gypsum content, among those studied. Leaf and soil materials from the sampled localities were analyzed for the 33 elements studied. The elements with concentration values < 0.01 were deleted. The BCF parameter was calculated for the different localities to obtain an average value for each plant species population.

### 3.7. Statistical Analysis

The statistical analysis of variance and Student’s *t* tests were performed using SPSS v29.0.2 software [128]. ANOVA and Sheffe’s post hoc tests were conducted to compare leaf concentrations in groups of taxonomical families and functional types (narrow gypsophile, wide gypsophile and gypsovag). In the case of the narrow gypsophile *G. arrosti*, field sampling allowed us to distinguish specifically populations of this taxon growing either on gypsum or limestone substrates. ANOVA tests were executed using the values obtained for the BCF of these species grouped according to soil type.

Several Student’s *t* tests were implemented to assess whether the grouping made by the succulence character held any difference between succulent- and non-succulent-leaved species among the element contents analyzed.

In addition, using Past v5 software [129], principal component analysis (PCA) was conducted to compare patterns in the chemical composition of the leaves, taking the order of variables for the sampled species as follows: Ca-S-Mg and N-P-K concentrations.

Finally, ternary plots were used as a visual representation of the multiclass classification of the leaf samples [55,130]. In the ternary charts, three variables were represented simultaneously, allowing for the representation of the relationships between N-P-K and Ca-S-Mg leaf contents.

## 4. Conclusions

Leaf samples of gypsophytes and gypsicolous soils have been collected and analyzed to obtain preliminary results about the nutritional content of gypsophile flora in Sicily. The presented outcomes indicate that gypsum habitats, due to their unique characteristics, are determinants of the nutritional behavior of plant species. There are some candidate elements that cause soil imbalances, but among them, Ca and Mg appear to always be critical [62,130]. In fact, gypsophile species efficiently accumulate Ca. Succulent-leaved species, such as *G. arrostii*, *Sedum*, and *Petrosedum*, showed high Ca concentrations, and their leaf Mg levels were also relatively high, as suggested by Moore et al. [88]. However, from the nutritional point of view, the concepts of ‘wide gypsophile’ and ‘narrow gypsophile’ could not fit perfectly with the studied situations. Conversely, the taxonomical family appears to have a great influence on the nutritional response of gypsophile plants to gypsum. In fact, there are families, such as Brassicaceae, that can accumulate high S concentrations in their tissues. Brassicaceae plant species are shown as accumulators, perhaps not for living on gypsum, but because of pre-adaptation, which gives them an advantage thanks to their ability to colonize gypsum soils. The Crassulacean are good accumulators of Ca; this element likely fulfills diverse physiological functions. Osmotic equilibrium/osmotic adjustment, water storage, solar protection, and other physiological functions would justify the high Ca concentrations in their tissues.

The genus *Gypsophila* includes plant species with the capacity to accumulate Ca and S. The species *G. arrostii* accumulates these elements at high concentrations. It appears that excess Ca and S can be stored in old leaves.

To summarize more explicitly the adaptive traits of the plants studied, we can conclude that succulent-leaved species accumulate significantly higher Ca concentrations, and narrow gypsophile species, in general, accumulate more S in their tissues. These facts would determine an advantage in terms of nutritional adaptation to succeed in the ecological conditions of gypsum soils.

BCF results proved to be very useful for characterizing the relationship between soil composition and the ability of plants to accumulate certain nutrients. These indexes support the bioaccumulation capacity of certain gypsophile species such as *G. arrostii* or *Diplotaxis harra* subsp. *crassifolia* for C, N, K, Mg, or Na. They also pointed to the accumulation of high Sr concentrations in leaves of the narrow gypsophile *Erysimum metlesicsii*, which may be a remarkable finding. In addition, a study on *G. arrostii* indicated that this plant tends to hyperaccumulate S when growing on limestone substrates, and this nutritional behavior would tend to compensate for the imbalance of elements in the soil.

Some of these species, for example, *G. arrostii* or *Diplotaxis harra* subsp. *crassifolia,* that are capable of accumulating certain elements in high quantities and that, in addition, have a wide distribution and grow quickly, could be useful in phytoremediation for the elimination of these elements from contaminated soils.

In conclusion, the nutritional mechanisms of plant species living on gypsum substrates deserve special attention because it will contribute to revealing the adaptive strategies that have allowed them to survive under extremely stressful conditions. The elemental characterization of the flora and vegetation on the gypsum outcrops of Sicily would contribute further support to highlight their uniqueness, in addition to their richness and endemicity. This would provide further justification for the conservation of these natural areas, as a priority target in global environmental policies and a key component of the UN 2030 Sustainable Development Agenda.

## Figures and Tables

**Figure 1 plants-14-00804-f001:**
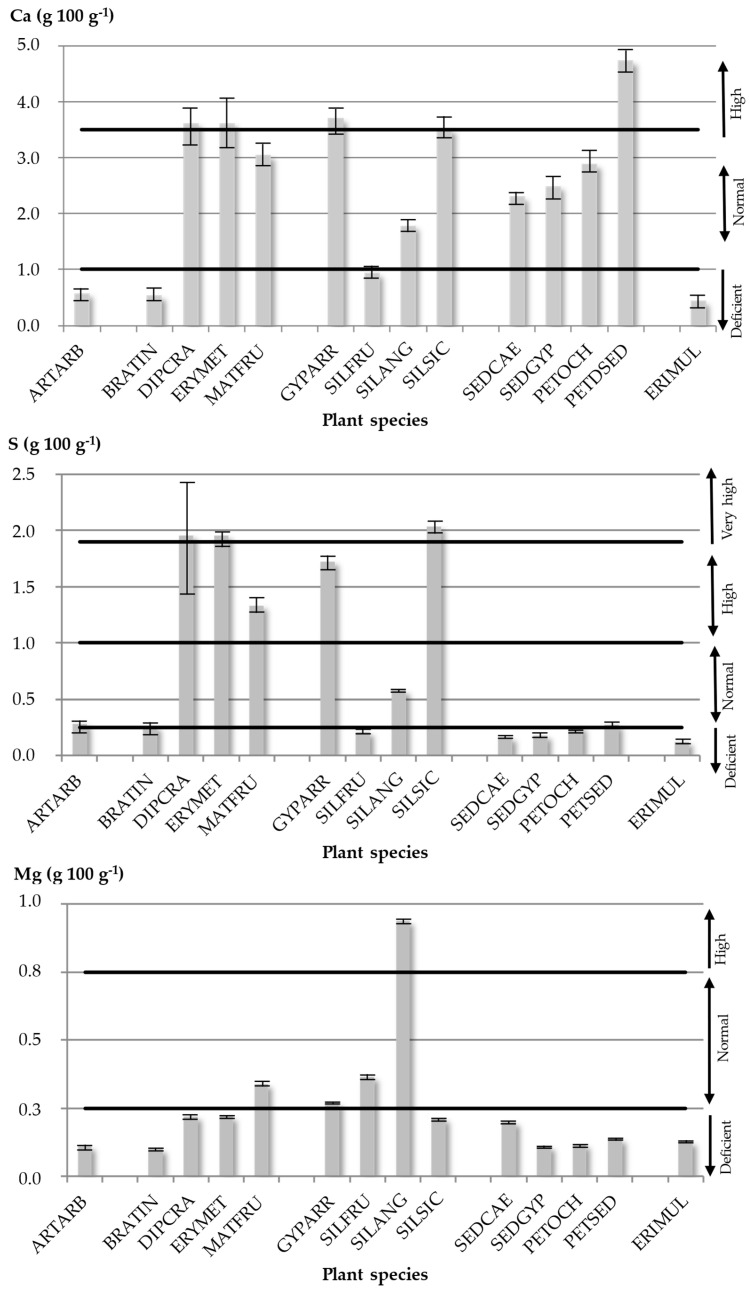
Average contents of Ca, S and Mg for foliar samples. The horizontal lines indicate the thresholds of these elements, following Merlo et al. [38].

**Figure 2 plants-14-00804-f002:**
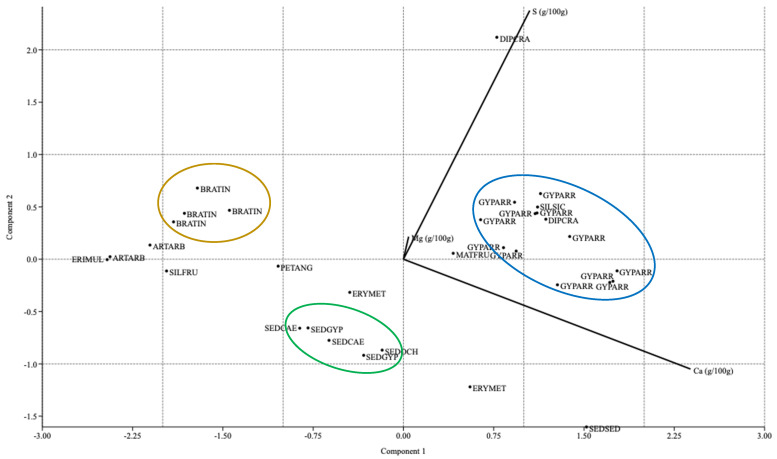
Principal component analysis. Ca, S, and Mg contents were used as ordering variables.

**Figure 3 plants-14-00804-f003:**
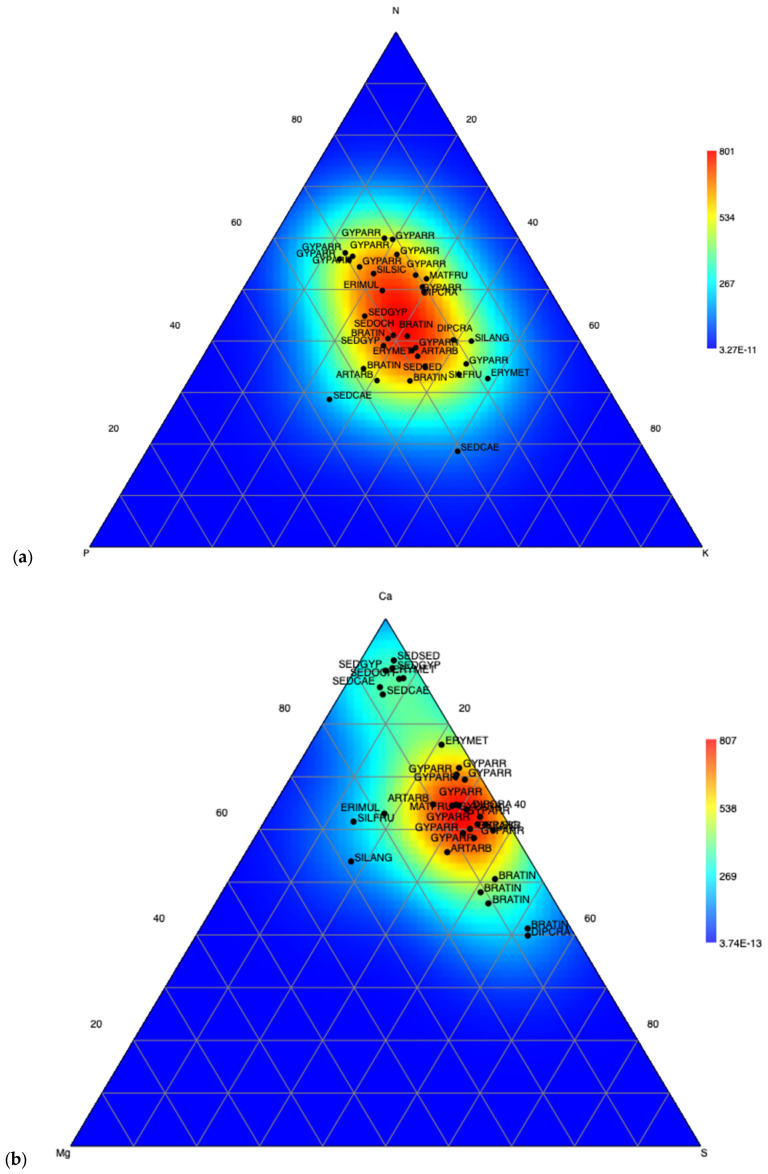
Ternary plot showing the stoichiometric relationship of N-P-K (**a**) and Ca-Mg-S (**b**) foliar contents in plant samples (g 100 g^−1^). For visual purposes, the P concentration was multiplied by a factor of 10.

**Table 1 plants-14-00804-t001:** Soil samples. Here is a description of the localities, dominant plant in the vegetation canopy, MGRS coordinates, altitude, percentage of gypsum, electrical conductivity, and pH.

Site	Dominant Plants	MGRS (ZONE 33S)		H (m)	% Gypsum	EC dS m^−1^	pH_W_
		X	Y				
Glibiscemi	*G. arrostii* subsp. *arrostii*	435419	4118582	440	78.33	2.78	7.84
Glibiscemi	*G. arrostii* subsp. *arrostii*	435511	4118308	519	44.52	2.82	7.90
Glibiscemi	*Sedum gypsicola* subsp. *trinacriae*	435509	4118310	517	55.91	2.94	8.00
Glibiscemi	*Sedum gypsicola* subsp. *trinacriae*	435419	4118582	438	72.46	2.97	7.91
Glibiscemi	*Quercus ilex*	435419	4118582	435	61.18	2.90	7.76
Realmonte	*Ampelodesmos mauritanica*	364177	4132388	220	44.16	2.03	8.10
Realmonte	*Erica multiflora* subsp. *multiflora*	364165	4132394	209	36.75	1.69	8.59
Realmonte	*G. arrostii* subsp. *arrostii*	364269	4132213	279	50.77	3.03	7.51
Castelmola	*G. arrostii* subsp. *arrostii*	523959	4191241	639	2.96	1.25	7.36
R. Entella	*G. arrostii* subsp. *arrostii*	333979	4182564	424	95.64	3.17	7.70

**Table 2 plants-14-00804-t002:** Average content of main mineral element from sampled soils.

Site	Macronutrients (g 100 g^−1^)								Micronutrients (mg kg^−1^)						Other Elements (mg kg^−1^)					
	C	N	Ca	Mg	K	P	S	Na	B	Mn	Ni	Zn	Cr	Co	Cu	Pb	Sr	Ti	Tl	V
Realmonte	9.80	0.40	12.87	0.52	0.29	0.05	1.50	0.03	17.52	281.24	11.36	28.02	18.15	3.25	9.24	16.33	526.85	136.97	10.02	31.33
R. Entella	12.09	0.15	4.58	0.32	0.16	0.01	3.22	0.04	9.34	105.56	9.25	24.39	6.51	3.41	4.95	18.19	341.61	137.17	1.36	9.05
Glibiscemi	3.29	0.13	8.80	0.18	0.24	0.02	3.92	0.01	12.73	381.96	14.92	29.34	13.37	3.01	10.62	9.48	208.17	102.27	7.33	21.30
Castelmola	5.36	0.09	9.47	0.36	0.65	0.04	0.13	0.04	24.58	163.52	20.84	47.79	26.68	2.63	20.91	18.81	237.06	64.98	13.34	39.47

**Table 3 plants-14-00804-t003:** Average contents of essential macronutrients from sampled plants (g 100 g^−1^) and N:P ratio.

Species	Functional Type	C	N	P	N:P	K	Ca	S	Mg
Artarb	Gypsovag	44.94	2.29	0.21	10.72	2.17	0.57	0.28	0.11
Erimul	Gypsovag	55.91	0.72	0.04	18.23	0.33	0.43	0.13	0.13
Petang	Gypsovag	38.19	2.50	0.11	22.60	2.65	1.77	0.58	0.94
Petsed	Gypsovag	37.62	0.76	0.06	12.62	0.80	4.72	0.27	0.14
Sedcae	Gypsovag	40.33	1.15	0.19	6.13	1.61	2.30	0.16	0.20
Silfru	Gypsovag	42.60	2.01	0.14	14.63	2.61	0.93	0.21	0.37
Silsic	Gypsovag	39.38	3.35	0.17	19.59	1.25	3.50	2.03	0.21
Bratin	Narrow gypsophile	40.99	1.92	0.16	11.76	1.55	0.91	0.87	0.17
Erymet	Narrow gypsophile	37.41	1.45	0.10	14.45	1.53	3.04	0.45	0.12
Gyparr	Narrow gypsophile	38.25	3.13	0.15	20.51	1.35	3.73	1.76	0.28
Dipcra	Wide gypsophile	34.19	2.56	0.12	22.19	1.88	3.07	2.65	0.35
Matfru	Wide gypsophile	37.77	2.24	0.08	27.46	1.25	3.04	1.33	0.34
Petoch	Wide gypsophile	41.20	0.67	0.05	13.77	0.47	2.87	0.26	0.12
Sedgyp	Wide gypsophile	42.27	0.80	0.06	12.72	0.50	2.48	0.18	0.11

**Table 4 plants-14-00804-t004:** BCF results from representative plants collected at the R. di Entella, M. Gibliscemi, Realmonte, and Castelmola sites.

			Macronutrients						Micronutrients					Another		
Species	Functional Type	Site	C	N	Ca	K	Mg	P	S	B	Cu	Fe	Na	Al	Cr	Sr
Bratin	Narrow gyps.	R. Entella	3.39	12.20	0.20	9.80	0.54	13.46	0.27	2.42	0.38	0.02	1.05	0.02	0.22	0.70
Dipcra	Wide gyps.		2.81	11.65	0.55	11.32	1.50	7.72	1.04	2.59	0.33	0.03	19.40	0.04	0.04	3.46
Erymet	Narrow gyps.		3.09	9.19	0.66	9.68	0.38	8.25	0.14	2.91	0.31	0.06	2.98	0.08	0.07	3.77
Gyparr	Narrow gyps.		3.12	16.73	0.70	17.23	1.27	14.15	0.57	3.00	0.61	0.02	2.15	0.01	0.04	0.87
Matfru	Wide gyps.		3.12	14.22	0.66	7.86	1.06	6.72	0.41	6.20	0.65	0.11	5.60	0.16	0.14	3.59
Sedcae	Gypsovag		3.34	7.33	0.50	10.14	0.63	15.49	0.05	2.13	0.84	0.12	2.11	0.16	0.17	0.97
Sedgyp	Wide gyps.		3.50	5.66	0.60	4.09	0.47	6.11	0.05	1.63	0.67	0.03	0.50	0.04	0.06	1.18
Petoch	Wide gyps.		3.41	4.25	0.63	2.98	0.36	4.00	0.08	1.63	0.46	0.02	0.35	0.03	0.03	1.90
Petsed	Gypsovag		3.11	4.81	1.03	5.08	0.43	4.94	0.08	2.34	0.71	0.02	0.78	0.04	0.14	3.36
Dipcra	Wide gyps.	Glibiscemi	10.43	25.51	0.41	8.04	1.23	6.09	0.50	1.24	0.25	0.01	28.70	0.01	0.04	2.25
Gyparr	Narrow gyps.		12.10	24.99	0.45	3.96	0.77	5.22	0.36	2.09	0.40	0.01	1.65	0.01	0.02	0.66
Sedgyp	Wide gyps.		12.82	5.51	0.25	1.45	0.40	2.30	0.05	0.90	0.24	0.01	2.09	0.02	0.02	0.38
Gyparr	Narrow gyps.	Realmonte	4.57	5.65	0.36	4.74	0.50	1.90	0.48	1.89	0.31	0.01	1.21	0.00	0.03	0.26
Gyparr	Narrow gyps.	Castelmola	7.13	36.79	0.40	1.67	0.75	3.67	14.06	1.54	0.21	0.00	0.50	0.00	0.03	0.36
Petang	Gypsovag		7.13	28.49	0.19	4.09	2.58	2.67	4.55	0.71	0.25	0.01	22.08	0.01	0.03	0.20
Silsic	Gypsovag		7.35	38.23	0.37	1.93	0.58	4.13	16.02	1.25	0.23	0.01	0.54	0.01	0.09	0.14

**Table 5 plants-14-00804-t005:** BCF results from *G. arrostii* sorted according to substrate type; *p*-values were obtained from ANOVA comparisons.

Site	Substrate	N	K	Na	S
**Castelmola**	Limestone	47.02	1.66	0.58	15.55
		40.79	1.71	0.43	12.34
		36.75	1.14	0.13	12.85
		39.93	1.41	0.31	12.20
		37.03	1.16	0.28	12.68
**Glibiscemi**	Gypsum	24.99	3.96	1.65	0.36
**R. Entella**		18.27	15.91	1.61	0.53
**R. Entella**		15.19	18.55	2.69	0.61
**Realmonte**		6.42	5.12	1.04	0.49
**Realmonte**		4.89	4.36	1.37	0.48
	*p*-value	0.00	0.01	0.00	0.00

**Table 6 plants-14-00804-t006:** Sampled plant species and acronyms. Taxonomical family. Functional type.

Plant Species	Abrev.	Family	Site	Succulent	Functional Type
*Artemisia arborescens*	Artarb	Asteraceae	R. Entella	No	Gypsovag
*Artemisia arborescens*	Artarb	Asteraceae	Glibiscemi	No	Gypsovag
*Brassica villosa* subsp. *tineoi*	Bratin	Brassicaceae	R. Entella	Yes	Narrow gyps.
*Diplotaxis harra* subsp. *crassifolia*	Dipcra	Brassicaceae	R. Entella	Yes	Wide gyps.
*Diplotaxis harra* subsp. *crassifolia*	Dipcra	Brassicaceae	Glibiscemi	Yes	Wide gyps.
*Erysimum metlesicsii*	Erymet	Brassicaceae	R. Entella	No	Narrow gyps.
*Matthiola fruticulosa* subsp. *fruticulosa*	Matfru	Brassicaceae	R. Entella	No	Wide gyps.
*Gypsophila arrostii* subsp. *arrostii*	Gyparr	Cariophyllaceae	R. Entella	Yes	Narrow gyps.
*Gypsophila arrostii* subsp. *arrostii*	Gyparr	Cariophyllaceae	Monte Conca	Yes	Narrow gyps.
*Gypsophila arrostii* subsp. *arrostii*	Gyparr	Cariophyllaceae	Castelmola	Yes	Narrow gyps.
*Gypsophila arrostii* subsp. *arrostii*	Gyparr	Cariophyllaceae	Glibiscemi	Yes	Narrow gyps.
*Petrorhagia illyrica* subsp. *angustifolia*	Petang	Cariophyllaceae	Castelmola	No	Gipsovag
*Silene fruticosa*	Silfru	Cariophyllaceae	R. Entella	No	Gypsovag
*Silene italica* subsp. *sicula*	Silsic	Cariophyllaceae	Castelmola	No	Gipsovag
*Petrosedum ochroleucum* subsp. *mediterraneum*	Petoch	Crassulaceae	R. Entella	Yes	Wide gyps.
*Petrosedum sediforme*	Petsed	Crassulaceae	R. Entella	Yes	Gypsovag
*Sedum caeruleum*	Sedcae	Crassulaceae	R. Entella	Yes	Gypsovag
*Sedum gypsicola* subsp. *trinacriae*	Sedgyp	Crassulaceae	R. Entella	Yes	Wide gyps.
*Sedum gypsicola* subsp. *trinacriae*	Sedgyp	Crassulaceae	Glibiscemi	Yes	Wide gyps.
*Erica multiflora* subsp. *multiflora*	Erimul	Ericaceae	Glibiscemi	No	Gypsovag

## Data Availability

The original contributions presented in this study are included in the article/Appendix A. Further inquiries can be directed to the corresponding author.

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
