# Peer review of "Elemental Screening and Nutritional Strategies of Gypsophile Flora in Sicily"

_plants, 2025, doi:10.3390/plants14050804_

Round 1

Reviewer 1 Report

Comments and Suggestions for Authors

Abstract: 

1. The authors used abbreviations without defining them. They should first provide the full terms, followed by the abbreviations in parentheses.

2. Line 14: 'Elsewhere' is not used appropriately, as it sounds too informal for the context. The sentence structure also needs improvement.

3. Line 17: The sentence is too long; try to simplify it to make it more understandable for readers.

4. Overall the abstract is poorly written and needs improvement.

Introduction:

1. The cited literature references are not in English, which makes it difficult to assess the originality of the statements.

2. Irrelevant literature has been provided. The authors need to mention studies that are relevant to the current research and highlight the purpose of the study, explaining why it is important.

Results and discussion:

1. The figures need improvement. The Y-axis scales have commas, and the legends should be placed on the Y-axis.

2. The data in the figures did not show any standard deviation or standard error bars.

3. The sentence structure is ambiguous, and informal language has been used. Too many details are provided in the results; the authors need to briefly explain the main findings.

Materials and Methods:

1. In sections 3.1 and 3.2, the authors need to briefly explain the content.

2. Which software was used for statistical analysis and figure preparation?

3. In the tables and figures, the data do not show any standard deviation.

4. How many replications were used for each plant species?

Comments on the Quality of English Language

The manuscript should be reviewed by a native English speaker to improve the quality of the language and correct grammatical mistakes. If possible, the authors should consider sending the manuscript for professional English editing services.

Author Response

Abstract: 

  1. The authors used abbreviations without defining them. They should first provide the full terms, followed by the abbreviations in parentheses.

Fixed, we want to thank the reviewer for this observation.

  1. Line 14: 'Elsewhere' is not used appropriately, as it sounds too informal for the context. The sentence structure also needs improvement.

Done.

  1. Line 17: The sentence is too long; try to simplify it to make it more understandable for readers.

Done.

  1. Overall the abstract is poorly written and needs improvement.

 Fixed, we want to thank the reviewer for this suggestion. Th abstract has been deeply modified.

Introduction:

  1. The cited literature references are not in English, which makes it difficult to assess the originality of the statements.

We appreciate the observation, but in order to frame the study of vegetation on Sicilian gypsum outcrops, we believe it is convenient to refer to studies in Italian.

  1. Irrelevant literature has been provided. The authors need to mention studies that are relevant to the current research and highlight the purpose of the study, explaining why it is important.

Thank you very much for this suggestion. Updated literature directly joined with the scope of the study have been included.

Results and discussion:

  1. The figures need improvement. The Y-axis scales have commas, and the legends should be placed on the Y-axis.

Done.

  1. The data in the figures did not show any standard deviation or standard error bars.

Fixed.

  1. The sentence structure is ambiguous, and informal language has been used. Too many details are provided in the results; the authors need to briefly explain the main findings.

Done. The results section has been modified and summarized.

Materials and Methods:

  1. In sections 3.1 and 3.2, the authors need to briefly explain the content.

Thank you very much. The content of these sections has been summarized.

  1. Which software was used for statistical analysis and figure preparation?

SPSS, Past and MS Excel software have been used for data analysis. This has been indicated in the text.

  1. In the tables and figures, the data do not show any standard deviation.

Done. We have included these items in figures. Moreover, SD are indicated in the extended analyses in the supplementary material.

  1. How many replications were used for each plant species?

Five replicates were taken per plant. This has been clarified in the text.

Comments on the Quality of English Language

The manuscript should be reviewed by a native English speaker to improve the quality of the language and correct grammatical mistakes. If possible, the authors should consider sending the manuscript for professional English editing services.

Done.

Reviewer 2 Report

Comments and Suggestions for Authors

This article examines the chemical composition of leaves and nutritional strategies of gypsophile plant flora in Sicily. The study presents important findings for the conservation of these habitats by revealing the ecological and nutritional strategies of plants adapted to gypsum substrates.

The article could be improved if some sections were added.

1- Abstract

Although the findings are specifically emphasized, practical implications could be emphasized more. For example, the implications for conservation strategies could be made more apparent.

2- Introduction

Information on plant communities in non-gypsophile habitats could be provided.

Information from similar studies conducted in different parts of Europe could be provided.

3- Methodology

The precision and measurement accuracy of soil data should be specified in more detail.

4- Discussion

The results can be discussed in a wider ecological and evolutionary context.

The implications for practical conservation strategies should be clarified.

In addition, if such studies exist in other regions, the discussion could be strengthened.

After these corrections, the article should be checked by a native English speaker. Also please review the attached file.

Author Response

This article examines the chemical composition of leaves and nutritional strategies of gypsophile plant flora in Sicily. The study presents important findings for the conservation of these habitats by revealing the ecological and nutritional strategies of plants adapted to gypsum substrates.

The article could be improved if some sections were added.

Thank you very much for your suggestions, they have all been taken into consideration. With these modifications we believe that the quality of the work has improved considerably.

1- Abstract

Although the findings are specifically emphasized, practical implications could be emphasized more. For example, the implications for conservation strategies could be made more apparent.

Fixed. The abstract has been modified.

2- Introduction

Information on plant communities in non-gypsophile habitats could be provided.

Done. Reference has been made to other types of communities developed on peculiar soils that are also being studied.

Information from similar studies conducted in different parts of Europe could be provided.

Done. Several published studies on this subject have been included on various gypsophilous plants. Line 98.

3- Methodology

The precision and measurement accuracy of soil data should be specified in more detail.

Fixed. We have included a clarification on the sampling methodology. Line 538.

4- Discussion

The results can be discussed in a wider ecological and evolutionary context.

Done.

The implications for practical conservation strategies should be clarified.

Fixed.

In addition, if such studies exist in other regions, the discussion could be strengthened.

Done.

After these corrections, the article should be checked by a native English speaker. Also please review the attached file.

Done.

Reviewer 3 Report

Comments and Suggestions for Authors

General Comments

This study focuses on plant communities within the gypsum ecosystems of Sicily, particularly investigating the nutritional adaptation strategies of plants growing on gypsum soils, especially gypsum-loving plants (gypsophytes). The results demonstrate that these plants adapt to nutrient-poor environments by accumulating key elements (such as calcium, magnesium, and sulfur) in their leaves, with significant differences in nutrient accumulation patterns among species. The study also reveals these plants' remarkable ability to absorb certain elements (e.g., sulfur and strontium), which not only sheds light on their ecological adaptations but also holds potential implications for environmental remediation. These findings underscore the importance of Sicily's gypsum ecosystem as a priority conservation habitat in Europe and provide new insights into plant adaptation strategies under harsh environmental conditions.

Main Comments

The abstract does not clearly highlight the novelty of the study. For example, while it mentions the accumulation of calcium and sulfur by gypsum plants, it does not elaborate on the ecological significance and uniqueness of these traits. It is recommended to emphasize the direct link between these traits and ecological adaptation.

The discussion section could be enriched by comparing the adaptation strategies of plants in other gypsum ecosystems, which would enhance the study's contextual background and depth of argumentation. This would better showcase the unique contributions of this research.

The conclusion should more explicitly summarize the plants' unique adaptive traits, such as significant calcium or sulfur accumulation. Strengthening the ecological or applied significance of key findings in the conclusion is advised.

Excessive use of tertiary headings detracts from the focus of the main content. Consider consolidating some headings to make the narrative more concise and the logical structure clearer.

Minor Comments

The references are outdated, lacking citations to recent advances in the field. It is recommended to include important recent literature.

The reference format is inconsistent. For instance, the year in line 671 is not bolded. Ensure uniform formatting throughout.

The table alignment is inconsistent. For example, the numbers in Tables S4 and S5 are right-aligned, while those in other tables are center-aligned. Standardize the formatting of all tables.

Some tables lack column headers, such as in Figure S2. Adding clear column headers would enhance readability.

The expression “(g 100g-1)” in line 489 is non-standard and should be revised to a proper unit format.

The font size in Figure 3 is too small. Increasing the font size would improve readability.

In line 22, the sentence “cate bioaccumulation of C, N, K, in species with a certain degree of foliar succulence such as……” contains an extra comma that should be removed.

The sentence “Thus, plant species that grow exclusively on gypsum outcrops have been called gypsophytes” lacks causal rigor and seems redundant. Consider removing or rewriting.

In line 475, “The samples were taken from superficial soils (15–25 cm deep, where was possible)” contains the phrase “where was possible,” which is unsuitable for academic writing. Replace with “where conditions allowed” or a similar expression.

Author Response

REFEREE 3

This study focuses on plant communities within the gypsum ecosystems of Sicily, particularly investigating the nutritional adaptation strategies of plants growing on gypsum soils, especially gypsum-loving plants (gypsophytes). The results demonstrate that these plants adapt to nutrient-poor environments by accumulating key elements (such as calcium, magnesium, and sulfur) in their leaves, with significant differences in nutrient accumulation patterns among species. The study also reveals these plants' remarkable ability to absorb certain elements (e.g., sulfur and strontium), which not only sheds light on their ecological adaptations but also holds potential implications for environmental remediation. These findings underscore the importance of Sicily's gypsum ecosystem as a priority conservation habitat in Europe and provide new insights into plant adaptation strategies under harsh environmental conditions.

Thank you very much to the reviewer for the words towards the manuscript, they are really gratifying.

Main Comments

The abstract does not clearly highlight the novelty of the study. For example, while it mentions the accumulation of calcium and sulfur by gypsum plants, it does not elaborate on the ecological significance and uniqueness of these traits. It is recommended to emphasize the direct link between these traits and ecological adaptation.

Done. The abstract section has been deeply modified.

The discussion section could be enriched by comparing the adaptation strategies of plants in other gypsum ecosystems, which would enhance the study's contextual background and depth of argumentation. This would better showcase the unique contributions of this research.

Done. The discussion has been enriched by adding new references and expanding some sections with more in-depth argumentation.

The conclusion should more explicitly summarize the plants' unique adaptive traits, such as significant calcium or sulfur accumulation. Strengthening the ecological or applied significance of key findings in the conclusion is advised.

Fixed. A paragraph has been included to refer to this conclusion and the ecological relationship with gypsum soil.

Excessive use of tertiary headings detracts from the focus of the main content. Consider consolidating some headings to make the narrative more concise and the logical structure clearer.

Done. We have unified some headings.

Minor Comments

The references are outdated, lacking citations to recent advances in the field. It is recommended to include important recent literature.

We appreciate the reviewer's comment, more current references on studies in the same line of research have been included, however, it has been decided not to remove any references as it is considered as a fair recognition to pioneer researchers of gypsicolous plants.

The reference format is inconsistent. For instance, the year in line 671 is not bolded. Ensure uniform formatting throughout.

Done. The References section has been revised according to the journal's standards.

The table alignment is inconsistent. For example, the numbers in Tables S4 and S5 are right-aligned, while those in other tables are center-aligned. Standardize the formatting of all tables.

Fixed

Some tables lack column headers, such as in Figure S2. Adding clear column headers would enhance readability.

Done

The expression “(g 100g-1)” in line 489 is non-standard and should be revised to a proper unit format.

Fixed

The font size in Figure 3 is too small. Increasing the font size would improve readability.

Done. The size of the figure has changed.

In line 22, the sentence “cate bioaccumulation of C, N, K, in species with a certain degree of foliar succulence such as……” contains an extra comma that should be removed.

Fixed

The sentence “Thus, plant species that grow exclusively on gypsum outcrops have been called gypsophytes” lacks causal rigor and seems redundant. Consider removing or rewriting.

Fixed

In line 475, “The samples were taken from superficial soils (15–25 cm deep, where was possible)” contains the phrase “where was possible,” which is unsuitable for academic writing. Replace with “where conditions allowed” or a similar expression.

Done

Round 2

Reviewer 3 Report

Comments and Suggestions for Authors

I think it can be accepted now.